# A first-principles phase field method for quantitatively predicting multi-composition phase separation without thermodynamic empirical parameter

Swastibrata Bhattacharyya [1], Ryoji Sahara[2] & Kaoru Ohno [1]

To design tailored materials, it is highly desirable to predict microstructures of alloys without empirical parameter. Phase field models (PFMs) rely on parameters adjusted to match experimental information, while first-principles methods cannot directly treat the typical length scale of 10 μm. Combining density functional theory, cluster expansion theory and potential renormalization theory, we derive the free energy as a function of compositions and construct a parameter-free PFM, which can predict microstructures in high-temperature regions of alloy phase diagrams. Applying this method to Ni-Al alloys at 1027 °C, we succeed in reproducing evolution of microstructures as a function of only compositions without thermodynamic empirical parameter. The resulting patterns including cuboidal shaped precipitations are in excellent agreement with the experimental microstructures in each region of the Ni-Al phase diagram. Our method is in principle applicable to any kind of alloys as a reliable theoretical tool to predict microstructures of new materials.

[1] Department of Physics, Yokohama National University, 79-5 Tokiwadai, Yokohama 240-8501, Japan. [2] National Institute for Materials Science (NIMS), 1-2-1 Sengen, Tsukuba 305-0047, Japan. Correspondence and requests for materials should be addressed to K.O. (email: ohno@ynu.ac.jp)

Microstructures involving precipitations, grain boundaries, dislocations, and other defects play a decisive role in many important properties of alloys such as ductility, plasticity, toughness, magnetism, oxidation- and heat-resistances. The quest of new materials with desirable properties requires microstructure engineering of materials by changing composition, morphology, pressure, and temperature as well as doping, casting, and forging. Since there are many degrees of freedom, it is highly desirable to find fundamental breakthrough toward the design of tailored materials. For this purpose, powerful computational techniques are required to predict microstructures. Since the length scale of microstructures is typically 1–100 μm and the number of atoms involved is $10^{11}$–$10^{17}$, first-principles methods such as density functional theory (DFT)[1] are not applicable. Phase field models (PFMs)[2–4] offer a promising computational tool to study such phenomena, where microstructures are described by order parameters. However, PFMs are purely empirical and adjustable parameters must be used so that the computational output matches the experimental one. Therefore, they are not widely accepted in industries[5].

In this paper we first introduce how to combine first-principles method and PFM, and solve the previous problems of PFMs relying on parameters. Although, a number of theoretical studies have been conducted using atomistic methods[6–8], they are limited, for example, to the level of the cluster variation method (CVM)[9], which relies on a particular crystal lattice and a particular alloy composition. A CVM-based phase field method has been also developed to study microstructure evolution in alloys[10,11]. In this method the alloy composition was kept fixed and the order–disorder phase transformation was treated at different temperatures. However, this method cannot discuss the entire phase diagram. Here, we propose a first-principles and non-parameter based PFM. The method can clearly discriminate the local composition and evolution mechanism of microstructures of alloys without any material parameter.

Although our method is in principle applicable to any kind of alloys, we demonstrate its ability by treating Ni–Al binary alloys as an example, which have attracted considerable attention for their excellent mechanical properties; very hard and good oxidation- and heat-resistances suitable for turbine disks and blades[12,13]. With varying Ni and Al compositions, they undergo many phase transformations. We will reproduce the experimental phase diagram[14] and show the time evolution of microstructures. We will focus on the ordered phases only, because the description of liquid and disordered phases requires to introduce another non-conserved order parameter.

In conventional PFMs, the order parameters are described by continuous functions of space and time. The free energy is defined as a polynomial of these order parameters and therefore are continuous too. These kind of representations do not take into account of the local structures and compositions, which are very important to find phases and microstructures in materials. By decomposing the space into a fine regular 3D grid of unit cells containing tetrahedron or other polygon (Fig. 1) and then taking the continuous limit where the lattice constant goes to zero, one can consider number density as the order parameter, which will include the atomic arrangement and local composition and eventually can construct a more effective free energy. In case of Ni–Al alloys, the four atomic sites of each tetrahedron can be filled with either Ni, Al, or vacancy, as shown in Fig. 1b. Thus, instead of continuous order parameter, one will obtain integer functions defined as the local composition of $Ni_nAl_m$: $\varphi_{Ni}$, $\varphi_{Al} =$ 0–1, 1–2, 2–3, 3–4, 4–5 for $n$, $m = 0$, 1, 2, 3, 4, and use cluster expansion theory[15–17] to determine the local energy from ab initio DFT. To include off-lattice effects, the potential renormalization theory can be implemented, where the atomic

displacement is renormalized in a length scale shorter than the lattice constant[18]. This theory is applicable at high temperatures where Einstein model is valid. Using this discretized free energy definition, together with the potential renormalization theory for the temperature effect, we simulate the evolution of microstructures and phases in Ni–Al alloy systems at various compositions. We fix the temperature at $T = 1027\,°C$ (=1300 K), which is typical in jet-engine turbines. Our results are of great match with the experimental and conventional phase field findings.

## Results

**The first-principles free energy and the diffusion equation.** The resulting local free energies $F(\varphi_{Ni}, \varphi_{Al})$ are shown in Figs. 2 and 3, and are plotted in 1D and 2D in Fig. 4a, b. Each block (plateau) in the 2D (1D) plot corresponds to an integer $(n, m)$ composition of the $Ni_nAl_m$ alloy, including vacancies for $0 \le (n + m) \le 4$ and interstitial atoms for $(n + m) > 4$. For $n + m = 5$, the most stable trigonal unit cell[19] is chosen with the same volume as the cubic unit cell. The explicit functional form for the free energy can be represented as:

$$F(\varphi_{Ni}, \varphi_{Al}) = \sum_{\substack{n,m \\ 0 \le (n+m) \le 6}}^{6} f_n(\varphi_{Ni})f_m(\varphi_{Al})E_{Ni_nAl_m}. \quad (1)$$

Where, $f_i(x) = \theta(x - i) - \theta(x - i - 1)$ and $E_{Ni_nAl_m}$ is the energy/renormalized energy of the $Ni_nAl_m$ cluster. The free energy is non-negative when $(n + m) > 6$ and increases sharply, which is described by a polynomial for $(n + m) < 0$ and $(n + m) > 6$ so as to avoid unrealistic compositions. The chemical potentials are given by

$$\mu_{Ni} = F(\varphi_{Ni} + 0.5, \varphi_{Al}) - F(\varphi_{Ni} - 0.5, \varphi_{Al}) - \varepsilon_{Ni}\nabla^2\varphi_{Ni}, \quad (2a)$$

$$\mu_{Al} = F(\varphi_{Ni}, \varphi_{Al} + 0.5) - F(\varphi_{Ni}, \varphi_{Al} - 0.5) - \varepsilon_{Al}\nabla^2\varphi_{Al}, \quad (2b)$$

where $\varepsilon_X$ ($X = Ni$ or $Al$) denotes the gradient energy coefficient of $\varphi_X$, and $\varepsilon_X\nabla^2\varphi_X$ represents the interface energy contribution. From the continuity equation $\partial\varphi_X/\partial t = -\nabla \cdot \mathbf{J}_X$ and the flux introduced by $\mathbf{J}_X = -M_X\nabla\mu_X$, the generalized Cahn–Hilliard equation is derived as

$$\frac{\partial\varphi_X}{\partial t} = M_X\nabla^2\mu_X. \quad (3)$$

Cahn–Hilliard equation is applicable to conserved order parameters such as the composition and the atomic density, while for non-conserved order parameters, Allen-Cahn equation should be used in the phase field model. We assume that $M_X$ and $\varepsilon_X$ are independent of the species $X$. Then, $M_X$ and $\varepsilon_X$ can be set arbitrary, e.g., at 0.00125 and 0.5, respectively, by rescaling time with $M_X$ and length with $\varepsilon_X$. Unlike atomistic simulations, in our coarse-grained model, there is only the change in the concentration and in this case to determine the time and length scales becomes particularly difficult. In order to avoid this difficulty, we assume simulation time and simulation cell size to be arbitrary units, which can be scaled to the experimental time and length if required. To obtain an exact relation between the simulation time (cell size) and the experimental time (length), one need to develop a method to calculate the mobility (interface energy) from first-principles and give this as an input parameter. But this generally requires a huge computation, and the method we are proposing here is to simply avoid this difficulty. Because the free energy is replaced with its local values, it is basically necessary to include the random force in the phase field equation. However, this does not affect the final pattern much. We have chosen the amplitude of the random force as 0.5 for all the calculations. For simplicity, we use a 2D model in a grid space of $124 \times 124$. The grid space

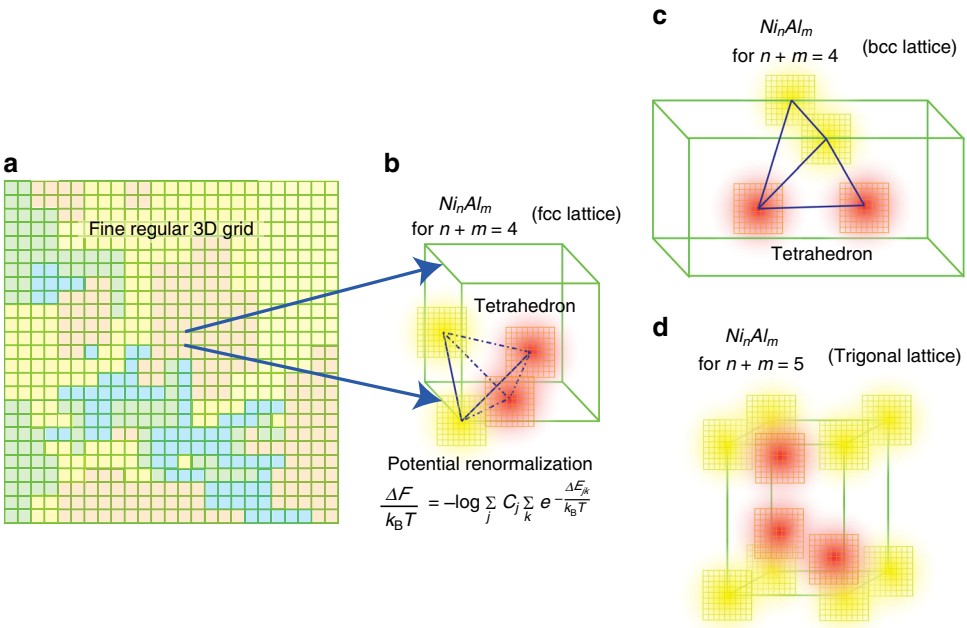

**Fig. 1** Pictorial representation of cluster expansion and potential renormalization theory. **a** The real space is divided into a fine regular 3D grid. One grid cube represents one unit cell and different colors represent different alloy composition within the unit cell. One unit cell of the Ni–Al alloy containing a tetrahedron formed by four nearest neighbor atomic positions for **b** fcc lattice, **c** bcc lattice. **d** Unit cell of trigonal structure. These atomic positions are occupied by $n$ Ni atoms (yellow) and $m$ Al atoms (orange) representing the composition $Ni_nAl_m$, in the cluster expansion theory. The 3D grids around each of the atoms are the points, where the atom was displaced during potential renormalization calculation

| Atomic structure | No vacancy | | | | |
|---|---|---|---|---|---|
| Alloy composition | $Ni_4$ | $Ni_3Al$ | $Ni_2Al_2$ | $NiAl_3$ | $Al_4$ |
| Total energy | −21.322 eV | −21.543 eV | −20.537 eV | −17.631 eV | −14.225 eV |
| Renormalization correction | 0.072 eV | 0.096 eV | 0.104 eV | 0.108 eV | 0.112 eV |
| Renormalized energy | −21.250 eV | −21.447 eV | −20.433 eV | −17.523 eV | −14.113 eV |
| Atomic structure | One vacancy | | | | Four vacancies |
| Alloy composition | $Ni_3$ | $Ni_2Al$ | $NiAl_2$ | $Al_3$ | Vacuum |
| Total energy | −13.852 eV | −14.127 eV | −12.896 eV | −10.475 eV | 0.0 eV |
| Atomic structure | Two vacancies | | | Three vacancies | |
| Alloy composition | $Ni_2$ | $NiAl$ | $Al_2$ | $Ni$ | $Al$ |
| Total energy | −7.099 eV | −7.515 eV | −6.030 eV | −1.717 eV | −1.895 eV |

**Fig. 2** Possible clusters and their energies in the cluster expansion theory (fcc lattice). Tetrahedral clusters with all the possible non equivalent combinations of Ni and Al, including vacancies at the lattice sites in a cubic unit cell. The total energies, calculated by ab initio DFT, renormalization corrections at $T = 1300$ K and the renormalized energies are shown below each structure of the no vacancy clusters, respectively. For the clusters with vacancy, total energies are shown. The Ni and Al atoms are represented by magenta and blue circles, while for vacancy cites white circles with dotted line are used

| | No vacancy | | | | |
|---|---|---|---|---|---|
| Atomic structure | | | | | |
| Supercell structure | | | | | |
| Alloy composition | $Ni_4$ | $Ni_3Al$ | $Ni_2Al_2$ | $NiAl_3$ | $Al_4$ |
| Total energy | −21.033 eV | −21.417 eV | −21.046 eV | −17.090 eV | −13.712 eV |
| Renormalization correction | 0.044 eV | 0.076 eV | 0.084 eV | 0.088 eV | 0.088 eV |
| Renormalized energy | −20.989 eV | −21.341 eV | −20.962 eV | −17.002 eV | −13.624 eV |

**Fig. 3** No vacancy clusters, and their energies in the cluster expansion theory (bcc lattice). Tetrahedral clusters formed by lattice points of two bcc cubic unit cells with all the possible non equivalent combinations of Ni (red sphere) and Al (blue sphere), without vacancies. The second row shows the corresponding $2 \times 2 \times 2$ supercells for the total energy calculation. The total energies per one tetragonal cluster, calculated by ab initio DFT, renormalization corrections at $T = 1300$ K and the renormalized energies are shown below each structure, respectively

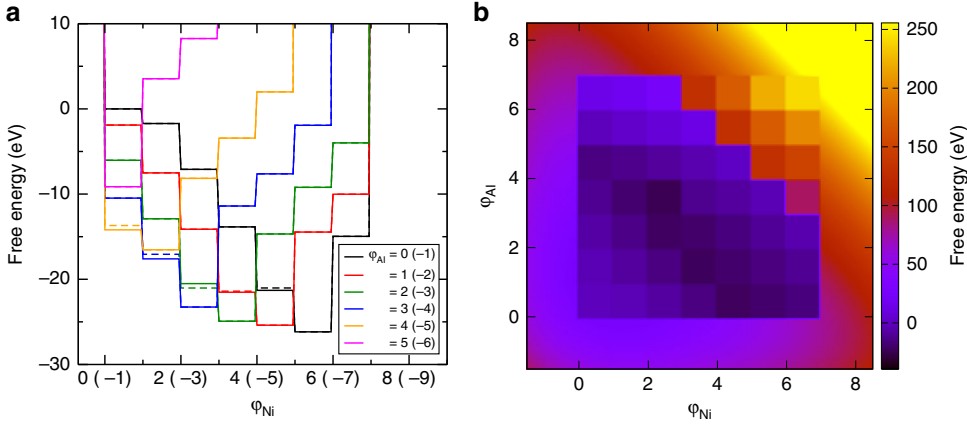

**Fig. 4** Free energy plot. **a** Plot of free energy versus $\varphi_{Ni}$ for $\varphi_{Al} = 0$ (black), 1 (red), 2 (green), 3 (blue), 4 (yellow), and 5 (magenta). **b** 2D plot of free energy as a function of $\varphi_{Ni}$ and $\varphi_{Al}$. The dotted lines in (**a**) represent the free energy values for the no vacancy clusters in the bcc lattice

($\Delta x$) and the time step ($\Delta t$) are set at 0.8 and 1.0, respectively. The PFM simulations are performed for a time step, until the microstructure becomes almost stable (typically around $t = 10^5$ for most of the compositions).

**Microstructure evolution at various alloy compositions**. Here, we restrict ourselves to the solid-state phases of $Ni_nAl_{4−n}$ with $n > 0.88$ (Ni > 22%) as shown in Fig. 5a. The initial structure contains Al- and Ni-rich seeds in a uniform matrix as shown in Fig. 5b. The resulting structures ($\varphi_{Ni}$) are plotted in Fig. 5c–t for the distinct eight solid phases at 1027 °C, corresponding to the number percentages of Ni as 22, 25.5, 30, and 35.5% for region I; 40% for region II; 41% for region III; 47, 52.5, and 60% for region IV; 65 and 70% for region V; 75% for region VI; 78, 79, 80, 82, and 84% for region VII, and 92% for region VIII. The resulting microstructures vary distinctively with the alloy composition for each phase region as explained below.

The variation in microstructures for region I ranging from Ni 22 to 38% is shown in Fig. 5c–f. For small Ni concentrations, the Ni-rich seeds start dissolving followed by Al-rich seeds forming a homogeneous solid solution with very small variation (Fig. 5c). In

the middle of this region, $Al_4$ precipitates are obtained within a matrix formed by mixed $Ni_2Al_3$ and $NiAl_3$. These precipitates were random with dendritic signatures for Ni 25.5% and rectangular for Ni 30% as shown in Fig. 5d, e. For snapshots of the simulation, see Fig. 6a. The Al-rich seeds grow and Ni-rich seeds dissolve with time, showing Ostwald ripening. The particle size decreases with increasing Ni concentration and the microstructure becomes almost homogeneous for Ni 35.5% (Fig. 5f) as we enter region II.

In the next region II ranging from Ni 38 to 40%, the initial structure disappears (Fig. 5g for Ni 40%) by Ni-rich seeds first getting dissolved followed by the Al-rich seeds. Figure 6b shows the snapshots. This region is the single phase region of $Ni_2Al_3$, where the total composition is $n + m = 5$.

In region III also, we obtained homogeneous pattern with some slight variation as shown in Fig. 5h for Ni 41%. The snapshots in Fig. 6c shows similar time evolution as Ni 40%.

Region IV is the widest among all solid phase regions. We observe uniform solid solution (Fig. 5i, j for Ni 47 and 52.5%) in this region except near the right boundary. Near the right boundary, Ni-rich seeds precipitate. First, circular shaped $Ni_3Al$ particles are formed in a matrix of $Ni_2Al_2$, which transformed

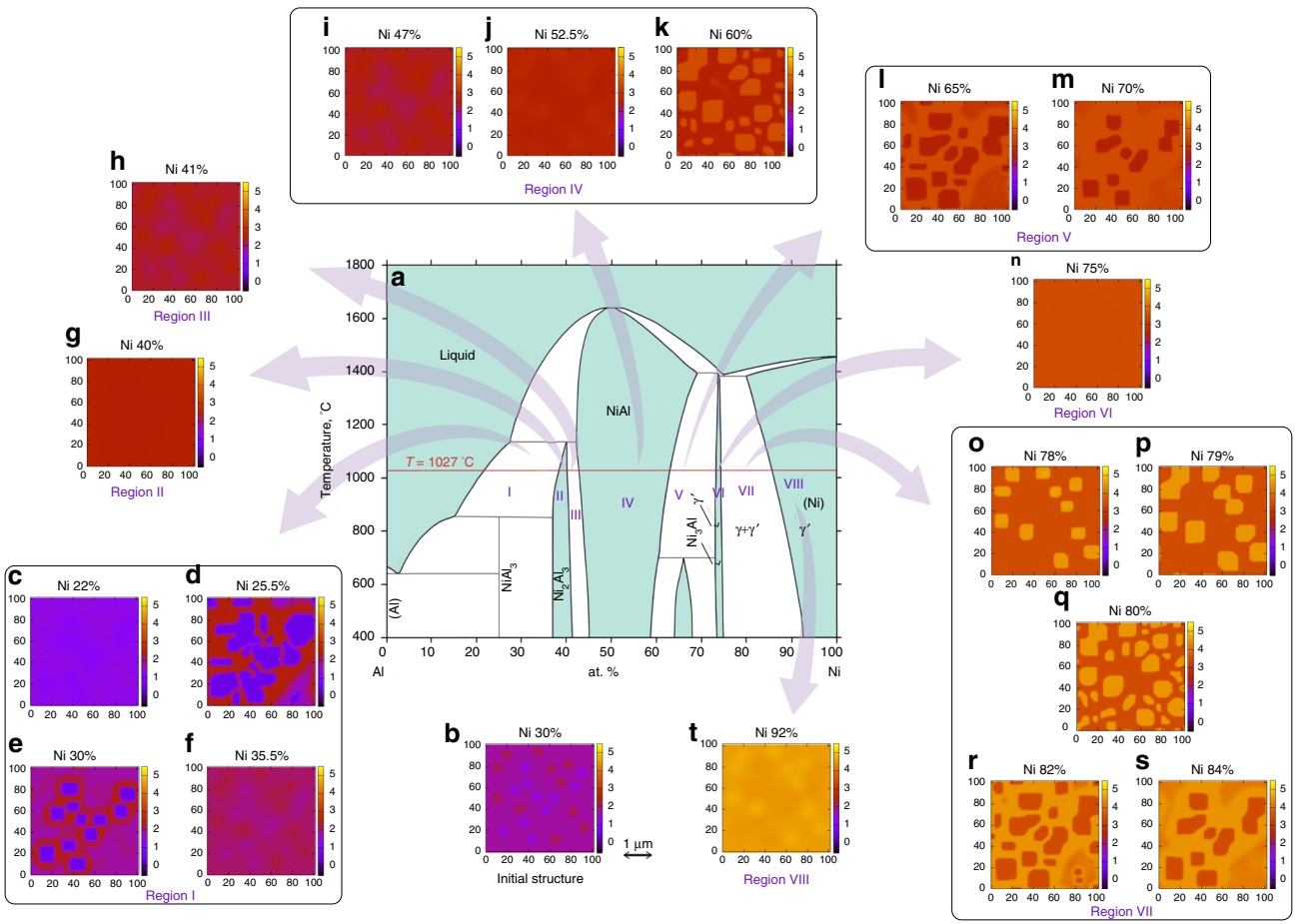

**Fig. 5** Microstructures at various alloy compositions. **a** Temperature–composition phase diagram for Ni–Al alloy[14]. The compositions at which phase field simulation results are shown falls on the $T = 1027$ °C line. **b** Plot of $\varphi_{Ni}$ at time $t = 0$ for composition $Ni_{1.2}Al_{2.8}$. Plot of $\varphi_{Ni}$, calculated using our phase field model after time step $t = 10^5$ ($t = 1.4 \times 10^5$ for (**o**, **p**) and $t = 3 \times 10^5$ for (**d**, **e**) are shown for the compositions **c** $Ni_{0.88}Al_{3.12}$, **d** $Ni_{1.02}Al_{2.98}$, **e** $Ni_{1.2}Al_{2.8}$, and **f** $Ni_{1.42}Al_{2.58}$ in region I; **g** $Ni_2Al_3$ in region II; **h** $Ni_{1.64}Al_{2.36}$ in region III; **i** $Ni_{1.88}Al_{1.12}$, **j** $Ni_{2.1}Al_{1.9}$, and **k** $Ni_{2.8}Al_{1.2}$ in region IV; **l** $Ni_{2.6}Al_{1.4}$ and **m** $Ni_{2.8}Al_{1.2}$ in region V; **n** $Ni_3Al$ in region VI; **o** $Ni_{3.12}Al_{0.88}$, **p** $Ni_{3.16}Al_{0.84}$, **q** $Ni_{3.2}Al_{0.8}$, **r** $Ni_{3.28}Al_{0.72}$, and **s** $Ni_{3.36}Al_{0.64}$ in region VII, and **t** $Ni_{3.68}Al_{0.32}$ in region VIII. The corresponding values of Ni % is written on top of each plot. The length scale, given here is a guideline for the readers to have a rough estimation and obtained by comparing with the microstructures as in ref. [23]

into square shapes at $t = 10^4$ as shown in Fig. 6d for Ni 60%. The particle size grows and coalesced particles are formed by some particles at $t = 6 \times 10^4$ along with the formation of some smaller particles in the matrix. With increasing Ni concentration, the particle size increases.

The microstructure composition changes in region V ranging from Ni 62 to 73% (Fig. 5l, m for Ni 65 and 70%). Here, Al-rich seeds precipitate as shown in Fig. 6e for Ni 65%. Spherical $Ni_2Al_2$ particles are formed within a uniform matrix of $Ni_3Al$ at $t = 4 \times 10^3$. They transform into rectangular shapes along with the formation of some coalesced ones ($t = 4 \times 10^4$). The particle size increases as the time proceeds ($t = 10^5$). The particle size decreases with increasing Ni concentration and distribution becomes homogeneous in the next region. Region VI is a single phase having no microstructure as shown in Fig. 5n for Ni 75%. The corresponding snapshots are shown in Fig. 6f.

In region VII, rectangular Ni particles are formed in the $Ni_3Al$ matrix from the Ni-rich seeds of the input structure. Internal structures appear in these particles as shown in Fig. 6g for Ni 78% at $t = 2 \times 10^3$. These particles grow further at later times to form square shapes composed of $Ni_4$ only ($t = 8 \times 10^4$). The particle size further increases and rectangular particles with various sizes are formed at $t = 1.4 \times 10^5$ (Fig. 5o). With increasing Ni concentration, the particle size increases and some coalesced

particles are formed (Fig. 5p). Some smaller particles with irregular shapes appear for Ni 80% (Fig. 5q). For Ni concentration higher than 80%, Al-rich seeds grow forming spherical shapes, that transform into rectangular shape as shown in Figs. 5r and 6h for Ni 82%. The resulting microstructure consists of $Ni_3Al$ particles embedded in pure $Ni_4$ matrix. For higher Ni concentrations, the particle size reduces and the microstructure disappears forming a uniform phase in the next region; see Figs. 5t and 6i for Ni 92%.

## Disscussion
To understand the growth mechanism of the microstructures from the initial pattern, we plot the change in local concentration and hence the local free energy at various time steps, as shown in Supplementary Fig. 1a, b for Ni 60%. We also plot the magnitude of the free energy gradient ($|\nabla F|$), which corresponds to the local stress of the system at each grid point in Supplementary Fig. 1c. As expected the local stress is non-zero only at the interface. The interfacial thickness is related to the (visible) non-zero gradient region. These plots show a sharp interface and we can expect bulk region even in the smallest precipitates of the microstructures. The plot of the total stress of the system versus the simulation time step shows an initial increase in stress until a maxima is

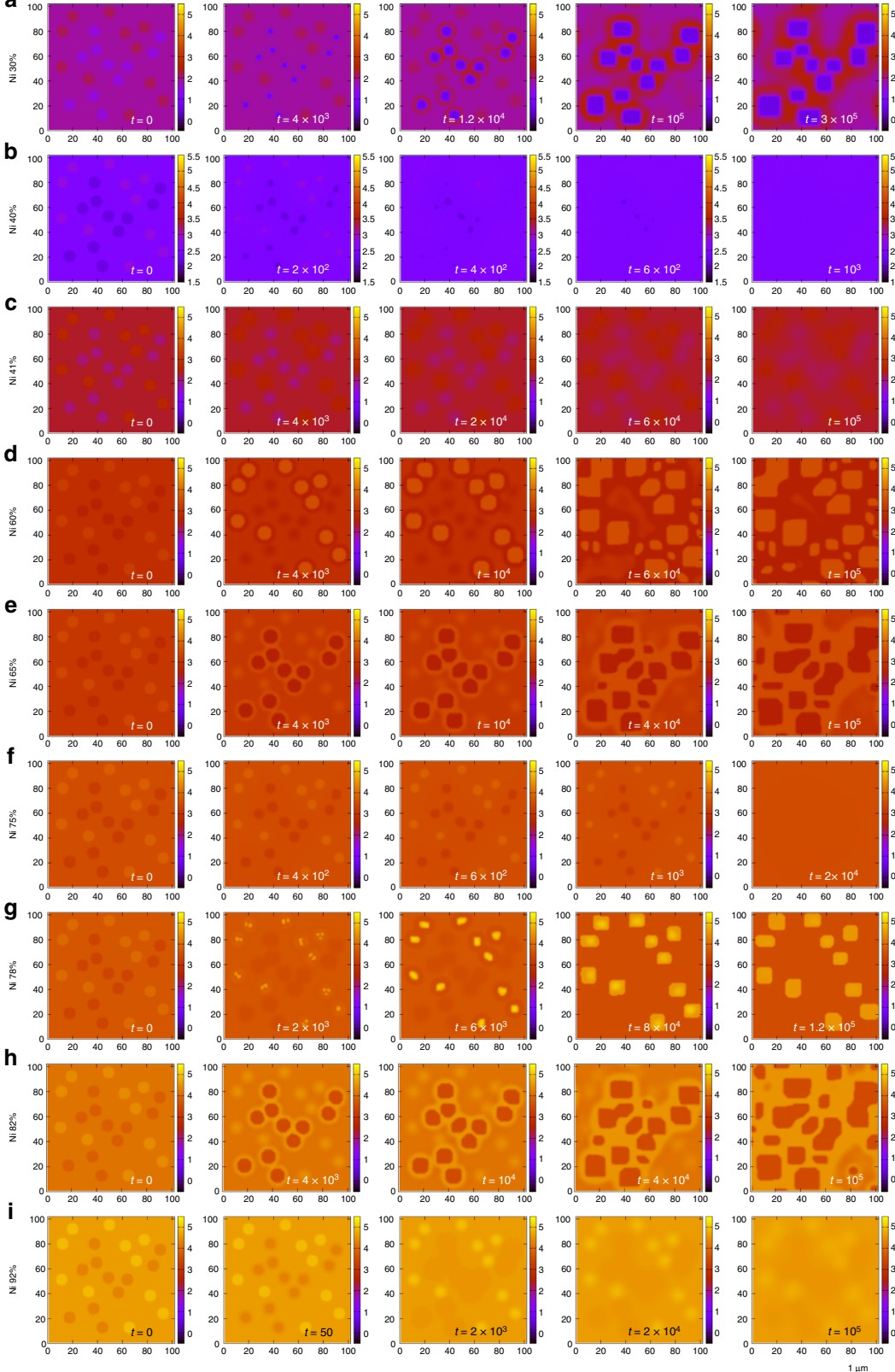

**Fig. 6** Time evolution of microstructure in various alloy compositions. Snapshots of phase field simulation for $\varphi_{Ni}$ at various times for **a** $Ni_{1.2}Al_{2.8}$, **b** $Ni_2Al_3$, **c** $Ni_{1.64}Al_{2.36}$, **d** $Ni_{2.4}Al_{1.6}$, **e** $Ni_{2.6}Al_{1.4}$, **f** $Ni_3Al$, **g** $Ni_{3.12}Al_{0.88}$, **h** $Ni_{3.28}Al_{0.72}$, and **i** $Ni_{3.68}Al_{0.32}$. The corresponding values of Ni % is written for each row and the simulation time is shown in the inset of each plot. The length scale, given here is a guideline for the readers to have a rough estimation and obtained by comparing with the microstructures as in ref. [23]

reached. After this increase, the stress decreases slightly and becomes constant after sufficient simulation time (see Supplementary Fig. 1d). This initial increase in the stress corresponds to the diffusion of the individual species. After the system forms a stable microstructure, the stress also becomes constant. In addition to the growth mechanism, this inherent local stress in particular at the phase boundaries influences the shape of the precipitates too and we have obtained cuboidal precipitates, which are peculiar feature of the NiAl alloy without explicitly introducing any external parameter for coherency stresses and/or strain. In atomic scale, the lattice mismatch is the basic origin of the coherency strain. The difference in the lattice parameter can be included in our PFM as vacancies and interstitials as well as the mixture of the two phases. This effect can be included in our PFM in which the free energy is a discrete variable for integer compositions. This is not possible in all previous PFMs, in which the free energy is a continuous variable with respect to compositions. This is a distinct advantage of the present PFM. The most important point is that, in our PFM, the density is automatically fine tuned by the clusters with vacancies and interstitials as well as the mixture of the phases. Therefore, the detailed lattice mismatch can be handled without introducing anisotropic elastic energies. Thus, we can conclude that the main origin of the cuboidal precipitates is that our free energy is a discrete variable. Indeed, if we change the definition of the free energy to be a continuous variable with respect to compositions, we obtain round precipitates.

The resulting microstructures and phases from our phase field method are listed in Table 1 for each region in the solid phase diagram and compared with the experimental and/or empirical phase field model results. In region VII, we show inversion of microstructure from $\gamma$-Ni precipitates in $\gamma'$-Ni$_3$Al matrix (Fig. 5o–q), to $\gamma'$-Ni$_3$Al precipitates in $\gamma$ matrix (Fig. 5r, s). However, there is not yet any experimental demonstration of microstructures having Ni precipitates in Ni$_3$Al matrix, commonly termed as inverse superalloy. However, there is indication of possibility of such microstructure in some experimental papers as these microstructures are expected to have better hardness compared to the normal alloys ($\gamma'$ precipitates in $\gamma$ matrix). Vogel et al.[20] showed $\gamma$-Ni particles in the $\gamma'$-Ni$_3$Al precipitates in hierarchical microstructures of Ni–Al alloy. A very recent paper[21] established an analogy between the inverse alloy and the hieirarchical Ni$_{86.1}$Al$_{8.5}$Ti$_{5.4}$ alloy. Therefore, we believe that the prediction of such an inverse alloy will be seen in experiments too.

Our results are everywhere in excellent agreement with the previously reported microstructures. We compare the rectangular particles in Fig. 5r with that of Fig. 1 (heat-treated at 1350 °C) of ref. [22], both for Ni 82%, in terms of its size, and estimated our length scale as $\Delta x \cong 0.03$ μm. With this value we estimate $\varepsilon_X$ to be

$8.9 \times 10^{-11}$ Jm$^{-1}$ (see Supplementary Note 1), which is comparable to the gradient energy coefficient used in the previous phase field calculation on Ni–Al alloys[23].

We have assumed fcc lattice for various compositions of the Ni–Al alloy system for our phase field calculation. The exceptions were for $n + m = 5$ cases and Ni$_2$Al$_2$ composition. For all the $n + m = 5$ compositions, we have used trigonal unit cell (similar to Ni$_2$Al$_3$ structure). Ni$_2$Al$_2$ on the other hand is known to have a bcc (CsCl) structure. For a more accurate calculation, we have modified the free energy by constructing a tetragonal cluster from the bcc lattice point as described by Allen et al.[24]. Unlike fcc lattice, the cluster is formed by two adjacent unit cells as shown in Fig. 3. We kept the volume of the unit cell containing one tetragonal cluster same as that of the fcc cluster such that the number density is preserved. For the free energy calculation, we constructed a $2 \times 2 \times 2$ supercell for each Ni–Al combination. The structures, total energy per tetragonal cluster, renormalization correction and the renormalized energies are shown in Fig. 3 for each no vacancy Ni–Al combination, respectively. The dotted plateaus in Fig. 4a denote the free energies for the clusters in a bcc lattice. As expected, the bcc energy is lower than the fcc energy for the Ni$_2$Al$_2$ composition only. We first repeated the first-principles phase field calculation for Ni 47, 52.5, and 60% using this free energy, and confirmed that the resulting microstructures are similar to the ones obtained with fcc lattice. Next, we replaced the fcc energy with the bcc energy only for the Ni$_2$Al$_2$ composition, and repeated more realistic phase field simulations shown here. In all regions of the phase diagram, the resulting microstructures using the bcc Ni$_2$Al$_2$ energy perfectly coincide with those using the fcc Ni$_2$Al$_2$ energy. This suggests the validity of using the fcc energy only. This is due to the fact that the difference between these two energies was occasionally small. Of course, it is better to choose the most favorable lattice having the lowest energy for each Ni$_n$Al$_m$ composition as in the present simulation. This example nicely demonstrates the validity of this treatment. The most important point is that the lattice (unit cell) can be different in different composition. Our method is a universal one and can be used for any alloys in any lattice structure.

Since a 3D model with those parameters determined by first-principles calculations can be used for constructing a more realistic model, we performed 3D simulations for some of the alloy compositions by taking a smaller system size of $40 \times 40 \times 40$ grid points, using the same simulation parameters as in the 2D simulations. The initial structure consists of two seeds in a uniform matrix of the alloy. The resulting microstructures (shown in Fig. 7) are very similar to the ones obtained in the 2D simulation even for the spatial scale of the resulting patterns. This suggests that the 2D simulations are good enough to reproduce the real microstructures. Using this model, we confirmed that the

**Table 1 Summary of the resulting microstructures/phases for various regions in the phase diagram**

| Region | Our results | Experiments |
|---|---|---|
| I | Al$_4$ precipitate in Ni$_2$Al$_3$ + NiAl$_3$ matrix | Ni$_2$Al$_3$, NiAl$_3$ and Al eutectic phase in Ni 31.5% Ni–Al alloy[29] |
| II, III | Single-phase region | Ni$_2$Al$_3$ single phase for Ni 40% at 800 K[30] |
| IV | Single-phase region | Single phase for Ni 45%[31] |
| | Right boundary: Ni$_3$Al particles in Ni$_2$Al$_2$ matrix | Ni$_3$Al microstructures in martensitic Ni$_2$Al$_2$ phase at 1150 and 1275 °C for 64.8 at% Ni–Al alloy[32] |
| V | Ni$_2$Al$_2$ particles in Ni$_3$Al matrix | NiAl particles in a matrix of Ni$_3$Al with some pure Ni phase for Ni 64.8% at 920 °C[32] |
| VI | Single-phase region | Ni$_3$Al single phase at Ni 75%[33] |
| VII | <Ni 80%: Ni particles in Ni$_3$Al matrix | Ni precipitation in the Ni$_3$Al matrix (empirical PFM)[34] |
| | >Ni 80%: Ni$_3$Al precipitates in pure Ni$_4$ matrix containing small Ni$_3$Al particles | $\gamma'$-Ni$_3$Al precipitates in the matrix of $\gamma$ phase, containing fine cuboidal $\gamma'$ particles for Ni 82%[22,35] |
| VIII | Single-phase region | Ni–Al solid solution at 570 K[36] |

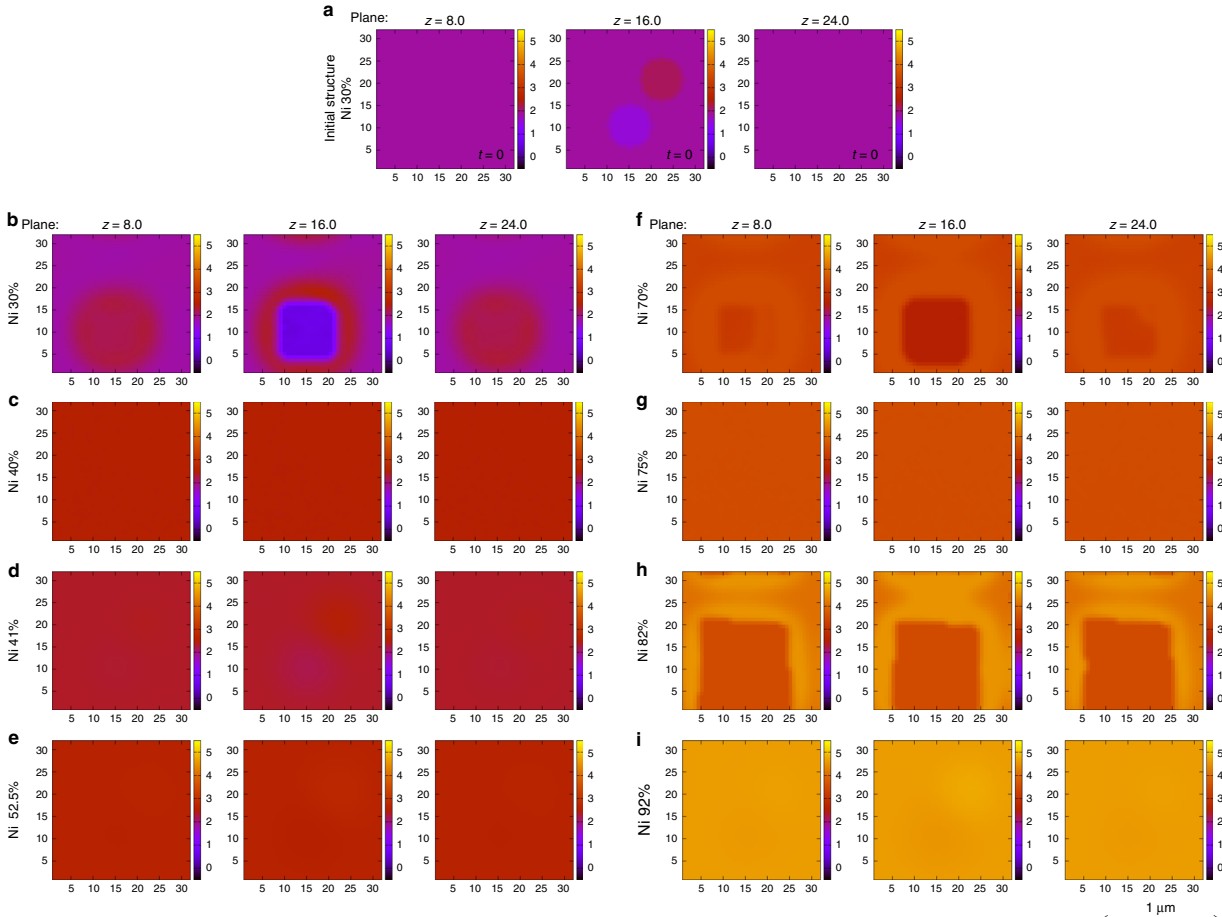

**Fig. 7** 3D simulations with (40 × 40 × 40) grid points and grid spacing of $\Delta x = 0.8$. **a** Plot of initial microstructure ($\varphi_{Ni}$) at planes $z = 8.0$, $z = 16.0$, and $z = 24.0$ for Ni 30%. Microstructures after a simulation step of $10^5$ for Ni concentration, **b** 30%, **c** 40%, **d** 41%, **e** 52.5%, **f** 70%, **g** 75%, **h** 82%, and **i** 92% are plotted for $\varphi_{Ni}$ at cross-sections along the vertical planes, $z = 8.0$, $z = 16.0$, and $z = 24.0$. The length scale, given here is a guideline for the readers to have a rough estimation and obtained by comparing with the microstructures as in ref. [23]

resulting PFM works well giving the reliable time evolution of microstructures for various compositions with no empirical parameter in the thermodynamic part of the model.

Thus, our first-principles phase field method is definitely a successful achievement in determining microstructures of the length scale of 1–100 μm purely from quantum mechanical ab initio theory. We strongly believe that the present method becomes a future theoretical standard for materials, which is fundamentally different from the existing methods limited only within explanation of experimental observations with empirical parameters. We are creating an automatic submission protocol for potential renormalization calculation that can be used to provide all the necessary input data for any system. This method has a potential to predict new useful materials in industries by supplying T–C phase diagram with microstructures, which is crucial to discuss about material properties in realistic applications. Our first-principles PFM is a universal method to perform large scale simulation for variety of materials at less computation time.

## Methods

**Coarse graining procedure.** In this model we have considered the number density $\varphi_X$ of the constituent elements, i.e., $X =$ Ni and Al as the conserved order parameters. We calculate the free energy, $F$ as a functional of these phase field variables by using cluster expansion theory[15–17]. It provides a very simple yet powerful approximation to calculate total energy of a system with a large number of substitutional structures. It allows us to calculate thermodynamic properties of a very large system by simplifying it into Ising-type models which deals with much

simpler and discretized coarse-grained systems. This method has been widely used in the calculations such as formation energies of random alloys and temperature–composition phase diagrams. For Ni–Al alloys, which are mostly found in an fcc structure, we use the tetrahedral approximation[15,16]. In a tetrahedron cluster (as shown in Figs. 1b, 2, and 3) there are maximum four possible atomic sites to be filled by the two atomic species Ni and Al or left vacant. In cluster expansion theory, the total energy can be expressed as a summation of the product of many-body interaction potentials ($J_i$) and multisite correlation functions $\xi_i$ for the $i$th order cluster ($i = 0, .., 4$ for tetrahedron approximation). The sum is over all the $i$th order clusters for this lattice type. Using DFT, the interaction potentials ($J_i$) and the total energy for various configurations $Ni_nAl_m$ ($0 \leq m + n \leq 4$) are calculated by filling the four sites of the tetrahedral unit cell by Ni or Al or vacancy. For Fe–Pt alloys with vacancies, see ref. [25]. In our model, there are total 20 such possible structures as shown in Figs. 2 and 3. For $n + m = 5$, a trigonal structure is chosen separately. The many-body interactions are evaluated by first-principles calculations based on density functional theory (DFT) using Vienna Ab initio Simulation Package (VASP)[26].

**Effect of temperature.** In the total energy calculations, the internal entropy arising by the displacement of the atoms in a length scale shorter than the lattice constant is very important in describing, e.g., nucleation, defects, crack propagation, and crystal growth. Without this effect, lattice models overestimate the order–disorder phase transition temperature. To include the internal entropy effect into our model, we apply potential renormalization theory[18], in which, the bare interparticle interaction in the off-lattice system is renormalized so as to give the same local partition function at a given temperature. This is done by discretizing the configuration space (Fig. 1b) and then taking the trace of the local movement of the atom from the lattice point for each of the configurations as follows:

$$\frac{\Delta F}{k_B T} = -\log \sum_j C_j \sum_k e^{-\frac{\Delta E_{jk}}{k_B T}}. \tag{4}$$

Here, $\Delta F$ denotes the change in the free energy due to the movement of the atom,

$\Delta E_{jk}$ is the increase in energy when one atom of the species $j$ = Ni or Al is moved to $r_i$, and $C_j$ is the concentration ratio. For normalization correction, Misumi's procedure [18] of dividing by 8 is inappropriate and a more suitable procedure is to divide by 4. The details about the potential renormalization calculation can be found in the Supplementary Note 2. Adding $\Delta F$ calculated in this way to the on-site energy, we obtain the required local free energy for each of the compositions shown in Figs. 2 and 3. The energies are renormalized at $T = 1300$ K (1027 °C). This procedure is valid at high temperatures, where the Einstein model becomes applicable. To partially incorporate the magnetism of Ni in the free energy, we performed spin polarized energy and renormalization (for non vacancy clusters) calculations for all the clusters containing Ni atom in VASP. The 1D and 2D plots of the free energy calculated in this way as a function of $\varphi_{Ni}$ and $\varphi_{Al}$ are shown in Fig. 2.

**Free energy functional**. In the previous PFMs, the free energy functional is expressed by a very simple polynomial of a continuous field variable[27]. In our PFM, the field variables $\varphi_X$ ($X$ = Ni or Al) have numbers varying from 0 to 5 such that $\varphi_X = 1 \sim 2$ corresponds to one site in the tetrahedron being occupied by $X$ atom, $\varphi_X = 2 \sim 3$ corresponds to two sites in the tetrahedron being occupied by $X$ atom, and so on. Because of this definition of $\varphi_X$, the condition for all sites of the tetrahedron to be filled by either Ni or Al is given by $\varphi_{Ni} + \varphi_{Al} = 5$. This is equivalent to the condition in terms of atomic fraction, i.e., $n + m = 4$ for the $Ni_n Al_m$ alloy. For a uniform definition, we normalize $\varphi_X$ to $x$ of $Ni_x Al_{4-x}$ as $x = \frac{4}{5}\varphi_X + (\frac{\varphi_X}{10} - \frac{1}{4})[\tanh 5(\varphi_X - 1) - \tanh 5(\varphi_X - 3.8)]$. This relation roughly produces $x = \frac{4}{5}\varphi_X$ for $x < 1.0$ and $x > 3.8$, and $x = \varphi_X - 0.5$ for $1.0 < x < 3.8$. Then our results can be compared with the experimental results, with any mixing composition of Ni and Al.

**Phase field simulation**. We have used this free energy, to perform phase field simulation at various regions in the phase diagram along the temperature line as shown in Fig. 3a. For the effect of temperature on the microstructures please see Supplementary Fig. 2 (for Ni 60% composition). For each of the global compositions $Ni_n Al_m$, we define the initial phase field densities, $\varphi_{Ni}$ and $\varphi_{Al}$ by giving a constant value each corresponding to $n$ and $m$, respectively, calculated using the expression relating $x$ to $\varphi_X$. In this uniform matrix we introduce some random circular seeds, half of them having composition $Ni_{n+c} Al_{m-c}$ and the remaining half as $Ni_{n-c} Al_{m+c}$, so that the total composition of the complete system remains the same as $Ni_n Al_m$ as shown in Fig. 3b for Ni 30% alloy. Here, $c$ is a very small constant parameter in our simulation (typically around 0.3). Defining this initial pattern, we perform the phase field simulation as per the above method. It is very important to input an initial pattern, without which the microstructures will not be formed. This is because a homogeneous pattern is a trivial solution to Eq. (3) corresponding to a stable distribution. To avoid the system going to such local minimum position, we need to assign some initial fluctuation in the input structure for example, by distributing random initial seeds. The amplitude $c$ of these seeds is of more importance than the number and distribution within the matrix. We observed that if $c < 0.3$, the system becomes homogeneous. This result strongly suggests the nucleation growth mechanism, which coincides with the experimental evidence of NiAl alloys as discussed by R. Moskovic[28].

## Data availability
All data generated and analysed during this study are included in the published article and its Supplementary Information.

## Code availability
Code that supports the findings of this study is available in the published article as Supplementary Information.

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

## Acknowledgements

We thank Yoshiyuki Kawazoe, Abhishek K. Singh, and Hannes Raebiger for carefully reading our paper. We are also very grateful to Riichi Kuwahara for creating an automatic submission protocol using Pipeline Pilot and CASTEP, with which we could perfectly confirm that all the results presented here are reproducible precisely. We have been indebted to the HPCI social and scientific priority issue "Creation of new functional devices and high-performance materials to support next-generation industries" to be tackled by using post-K computer promoted by MEXT for the use of the supercomputer facilities at the Institute for Solid State Physics, the University of Tokyo, at Hokkaido University, and at the Institute for Materials Research, Tohoku University (Project IDs. hp170268, hp170190, hp180125, and hp180220).

## Author contributions

K.O. designed the research. S.B. and K.O. wrote the paper and developed the code. S.B. performed the PFM calculation. S.B. performed the total energy calculations of some clusters and R.S. performed potential renormalization calculations. S.B., R.S., and K.O. analyzed the results.

## Additional information

**Competing interests:** The authors declare no competing interests.

