## [Peer Review File · Nature Communications]

Reviewers' comments:

Reviewer #1 (Remarks to the Author):

The authors present indeed a very interesting theoretical study featuring phase-field modeling of the Ni-Al binary system with parameters obtained from quantum-mechanical (ab initio) calculations. Well, it is indeed a great achievement. No doubt about that. But while the performance of the approach is very impressive and the use of quantum-mechanical calculations to determine the parameters is innovative, I do have a few questions related to the major claims that the approach is general (and may be used for other systems) and that all simulations are done without any empirical input.

First, it seems to me that the final microstructures would not form without those seeds that are distributed by the authors. What are the rules for their distribution? And how are these rules (if any) related to experiments? Second, what are the energy barriers for the diffusion of the individual species and what is the concentration of vacancies mediating the diffusion? Third, do I understand it well that the phase diagram in Figure 5 is, in fact, not computed by the authors? Well, does it mean that this approach can be applied only to those systems in case of which the phase diagram is known from experiments (including the structure of all phases which appear in the phase diagram)? Next, considering that Ni is magnetic, how is magnetism handled in these simulations and/or in VASP calculations and what is the corresponding magnetic entropy? Further, when the free energy is renormalized (what is a great method), what is the relation between the simulated time and the real one in experiments (when the discussed microstructures are being formed)? Well, when renormalizing and using dimension-free parameters, do I understand it well that the length scale is, in fact, introduced by using the size of morphological features from experimental microstructures? And lastly, there are indeed many different temperatures of experimental microstructures compared with simulations performed for a single temperature – why was this particular temperature of simulations chosen and could it be possible to have the experimental data for the simulated temperature?

When the authors address the above summarized questions, the manuscript may be considered for publication.

Reviewer #2 (Remarks to the Author):

The paper claims to present a novel method to predict microstructures in alloys from ab-initio calculations. The method is based on the Cahn-Hilliard equation, which the authors of the paper call the phase-field method. This is common in some part of the phase-field community, but not generally accepted.

Overall the paper presents interesting results. Microstructures are observed for different part of the composition-space, it is however difficult to evaluate how realistic the microstructures are. Aspects of nucleation of phases are not taken into account.

It is not clear to me why most of the microstructure show cuboidal precipitates, despite coherency stresses and strain not being taken into account. The authors might want to consider this in their discussion.

Overall calculations domains in the simulations are quite small, with rather coarse discretisation (few grid points). Given the diffuse interface in the CH/PFM used, I wonder how much bulk is present in the phases, or whether the microstructure consist mainly of interface region.

Region VII in Figure 5 shows an inversion of the microstructure from Y' in Y matrix to Y in Y' matrix. This is experimentally not commonly observed. Sensitivity of the model to changes in alloy

composition does not seem very high, as phase fraction do not seem to differ very much between results. As the authors stress out repeatedly that the model aims at supporting alloy development from ab-initio, this does matter.

Finally, the author claim repeatedly that the model has no empirical parameters. The chemical potential contains however the gradient energy term, the coefficient of which determined interface thickness and - more importantly - interfacial energy. Equation 2 contains a mobility coefficient M_x , which can be linked to the diffusion coefficient (through the atomistic mobility). These parameters were NOT determined from the ab-initio calculations, but chosen at random. Characteristic length scale, growth kinetics and resulting microstructure critically depend on these parameters. I do therefore NOT agree that there are no empirical parameters in the model. The authors should at least be more specific in their claim and state that the thermodynamic part of the model does not contain empirical parameters.

Finally, there have been a string of publication from T Mohri, dating as far back as 2004, combining phase-field with CVM calculations, producing microstructures from ab-initio. I would like the author to comment in their paper how their approach is different and novel.

Generally, the paper is well written and readable and contains interesting thoughts.

Reviewers' comments:

Reviewer #1 (Remarks to the Author):

The authors present indeed a very interesting theoretical study featuring phase-field modeling of the Ni-Al binary system with parameters obtained from quantum-mechanical (ab initio) calculations. Well, it is indeed a great achievement. No doubt about that.

We thank the Reviewer for finding our work interesting and considering it to be a great achievement.

But while the performance of the approach is very impressive and the use of quantum-mechanical calculations to determine the parameters is innovative, I do have a few questions related to the major claims that the approach is general (and may be used for other systems) and that all simulations are done without any empirical input.

First, it seems to me that the final microstructures would not form without those seeds that are distributed by the authors. What are the rules for their distribution? And how are these rules (if any) related to experiments?

The Reviewer is right that the microstructures will not be formed without any initial seeds. This is because, a homogeneous distribution corresponds to the trivial solution of the time evolution equation (Eq. 2) for the order parameter. This means, the right hand side of the equation will become zero corresponding to a stable distribution. This situation doesn't change even if we add a random force. To avoid the system going to such local minimum position, we need to assign some initial fluctuation in the input structure by distributing random initial seeds. The amplitude of these seeds is of more importance than the number and distribution within the matrix. We checked that if the amplitude of the seeds is less than 0.3 from that of the matrix, then the system becomes homogeneous. This result strongly suggests the nucleation growth mechanism, which coincides with the experimental evidence of NiAl alloys as discussed by R. Moskovic, J. Mat. Sci. 12(9) 1895-1902 (1977). We add this explanation in the revised manuscript (para 1, page 15).

Second, what are the energy barriers for the diffusion of the individual species and what is the concentration of vacancies mediating the diffusion?

Yes, this is a very important question. In atomistic description, a phase change occurs by the vacancy mediating atomic diffusion through energy barrier. However, in our coarse grained model, there is only the change in the concentration. Of course the average concentration is kept constant, and only local concentration can change from time to time. This can be mediated by vacancies in actual situation, but in our scheme, this process is not so clear. Probably, the most appropriate answer to this question is to present and discuss the local stress, calculated in our method. Since our method enables us to compute local free energy time to time and we can look at the change in the free energy distribution together with the change in the microstructure. Of course, final free energy distribution is almost

homogeneous except for the domain boundary, where very slight amount of the free energy gradient exists. Here, we show some snapshots of temporal evolution of the 2D map of the free energy and the free energy gradient for Ni-60% composition together with the temporal evolution of the microstructure:

Plot of magnitude of gradient of free energy as a function of simulation step

As seen in the plots, the sum of $|\nabla F|$ at all the grid points (which corresponds to the stress), increases initially, reaches a maxima and then gets saturated. This initial increase of stress corresponds to the diffusion of the individual species. After the system forms a stable microstructure the stress also

becomes constant. We add this explanation (para2, page 8) together with these figures in Supporting Information (Supplementary Fig. 1).

Third, do I understand it well that the phase diagram in Figure 5 is, in fact, not computed by the authors? Well, does it mean that this approach can be applied only to those systems in case of which the phase diagram is known from experiments (including the structure of all phases which appear in the phase diagram)?

The Reviewer is right that the phase diagram in Figure 5 is an experimental one (ref: National Institute for Materials Science (NIMS) AtomWork.<http://crystdb.nims.go.jp>.) and not computed by us. It is true that having a previous knowledge (from either experiments or theoretical calculation) about the phase diagram and the structures in each region in the phase diagram makes it easier to generate the clusters in the cluster expansion theory and calculate their free energy. However, even if we do not have any knowledge about the structural properties of material, we can still use standard density functional theory to determine the optimized structure and the lattice parameter at each composition. Please note that all the phase boundaries, which precisely coincide with experimental phase diagram, are obtained purely from our simulation results not by referring to the experimental phase diagram at all.

Next, considering that Ni is magnetic, how is magnetism handled in these simulations and/or in VASP calculations and what is the corresponding magnetic entropy?

We understand the concern by the Reviewer about the effect of magnetism of Ni and the resulting magnetic entropy on the microstructures. However to treat any magnetism or magnetic entropy we need to define an additional order parameter. This will further make our calculations more complicated. Therefore, we did not include the entropy term including the magnetic entropy in our model. However we incorporated the effect of magnetism in the free energy calculations of the clusters by performing spin polarized energy and renormalization (for non vacancy clusters) calculations for all the clusters containing Ni atom in VASP. We add this explanation in the Method section in the revised manuscript (para 1, page 14).

Further, when the free energy is renormalized (what is a great method), what is the relation between the simulated time and the real one in experiments (when the discussed microstructures are being formed)?

We completely agree with the Reviewer, that it is important to have an estimation about the time in the real scale taken by the alloy to form the respective equilibrium microstructures at each concentration. In the atomistic simulation, if it is possible, the diffusion time would become of course the real time. However, in our coarse grained model, there is only the change in the

concentration as we wrote in our reply to the previous comment on the diffusion of individual particles. In this case, to determine the time scale becomes particularly difficult. In order to avoid this difficulty, we assumed that the mobility (M) is unity and dimensionless quantity. Then the simulation time can be seen as having a unit of the mobility. That is, our simulation time is arbitrary unit, which can be scaled to the experimental time if required. To obtain an exact relation between the simulation time and the experimental time, one need to develop a method to calculate the mobility from first principles and give this as an input parameter. But this generally requires a huge computation, and the method we are proposing here is to simply avoid this difficulty. We have simulated each of the compositions for a sufficient time steps until a stable microstructure is formed in each composition and reported the microstructures that did not change much for any further simulation step. Since this was not clearly written in the original manuscript, we add this in the revised manuscript (para2, page 5).

We thank the Reviewer for considering the renormalization method, which we used to incorporate the temperature into the simulation, as being a great method. This method combines both the statistical and the quantum *ab-initio* method to calculate the free energy correction at any temperature. We assumed that the temperature of this renormalization method corresponds to the experimental temperature well within the allowed limit.

Well, when renormalizing and using dimension- parameters, do I understand it well that the length scale is, in fact, introduced by using the size of morphological features from experimental microstructures?

The Reviewer is right that we have in fact used dimension-free parameters for the interface energy (ϵ_x), which is related to the length scale for the microstructures. Thus our length scale is actually arbitrary. This is to avoid the huge computation of the interface energy to determine the length scale. We add this explanation together with the similar explanation for the time scale mentioned above in the revised manuscript (para 2, page 5). We compared the size of one of our microstructure to the experimental microstructure for the same composition and calculated the corresponding value of the surface energy (in our simulation) to check whether it is within the range of the experimental and/or the empirical phase field model value. Our ϵ_x value is on the same order of the one (the gradient energy coefficient) used in the PFM model by Ref. 19 (Zhu, J. et al. *Three-dimensional phase-field simulations of coarsening kinetics of γ' particles in binary Ni-Al alloys. Acta Materialia* 52, 2837–2845 (2004)). The length scale that we have calculated from this comparison is given in Figures 5, 6 and 7 as a guideline for the reader to have a rough estimation. We have now modified the figure captions for Figures 5, 6 and 7 to include this information.

And lastly, there are indeed many different temperatures of experimental microstructures compared with simulations performed for a single

temperature – why was this particular temperature of simulations chosen and could it be possible to have the experimental data for the simulated temperature?

This is an important point. As noted by the Reviewer, we have compared simulation results obtained at T=1300K to experimental results obtained at slightly different temperatures. As the number of variables for the different points in the Ni-Al composition-temperature (C-T) phase diagram is very large, for a systematic study, we have assumed a constant temperature (1300K) and varied the composition. Changing composition of the alloy is easier as we can use the same set of free energy values for that particular temperature. The reason we selected this temperature (1300K), is because, this is typical in jet-engine turbines. However we did not get experimental data to verify for all the alloy compositions at 1300K. For example, if we change the temperature, we can obtain the microstructures at different temperatures as shown below for Ni 60% composition:

With increase in temperature the bright phase is decreasing slightly. We have added this figure in the supplementary material (Supplementary Fig. 2) and write the comparison in its caption. We also briefly referred to this figure in the manuscript (para3, page 14). We took the advice of the Reviewer and now performed calculations at the temperatures of our experimental references and modified the justification letter with the microstructures corresponding to temperatures of the experimental reference.

When the authors address the above summarized questions, the manuscript may be considered for publication.

We thank the Reviewers for their constructive criticism and valuable advices and suggestions, which helped in improving the manuscript. We have addressed all the queries by the Reviewer and modified the manuscript accordingly. We believe that the Reviewers find the present version of the manuscript acceptable for the publication.

Reviewer 2

Reviewer #2(Remarks to the Author):

The paper claims to present a novel method to predict microstructures in alloys from ab-initio calculations. The method is based on the Cahn-Hilliard equation, which the authors of the paper call the phase-field method. This is common in some part of the phase-field community, but not generally accepted.

We thank the Reviewer for highlighting our method as a novel method to predict microstructures in alloys from ab-initio calculation and comparing this method to the conventional phase-field method, which uses empirical parameters in the free energy functional. The Reviewer is right that the Cahn Hilliard equation is not always acceptable. It can be applied to the conserved order parameters such as the composition and the atomic density, while for the non-conserved order parameter, Allen-Cahn equation should be used in the phase-field model. We shortly add this explanation in the revised manuscript (para 2, page 5).

Overall the paper presents interesting results. Microstructures are observed for different part of the composition-space, it is however difficult to evaluate how realistic the microstructures are. Aspects of nucleation of phases are not taken into account.

We thank the Reviewer to show his/her interest in our results and stating that the microstructures are observed for different part of the composition-space using our method. Yes, we can only compare our results with preexisting experimental data and other empirical phase-field simulations. The Reviewer is highlighting an important point that the nucleation of phases are important for the evolution of microstructure. We have included this effect by adding the random seeds in the initial structure. If we choose the homogeneous initial pattern without initial seeds, no change happens in the result. Only when we put initial seeds, non-trivial results come up. This means that the microstructures appearing in our simulation are all based on the nucleation growth mechanism. We shortly add this explanation in the revised manuscript (para 1, page 15).

It is not clear to me why most of the microstructure show cuboidal precipitates, despite coherency stresses and strain not being taken into account. The authors might want to consider this in their discussion.

The Reviewer is right that we have obtained cuboidal precipitates without including any external parameter for coherency stress and/or strain in our method. Cuboidal precipitates are peculiar feature of the NiAl alloy and the results coincide with many experimental findings. In our separate simulations, we have confirmed that the cuboidal shapes change to oval shapes when we

introduce Ti impurities, for example. In our simulation, the stress is automatically included due to the local variation of free energy. Thus there is inherent local stress in particular at the phase boundaries and we haven't explicitly introduced any external parameter for this in the system. This is an advantage in our method, as it limits many external parameters related to many physical and/or chemical effects including the stress. To see the stress, we plot the spatial profile of the gradient of the free energy density, which corresponds to the local stress. Some snapshots are shown as follows for Ni-60% composition:

Plot of magnitude of gradient of free energy as a function of simulation step

As seen in the plots, the sum of $|\nabla F|$ at all the grid points (which corresponds to the stress), increases initially, reaches a maxima and then gets saturated.

This initial increase of stress corresponds to the diffusion of the individual species. After the system forms a stable microstructure the stress also becomes constant.

Owing to these inherent local stresses, the NiAl alloy has a tendency to have cuboidal precipitates. We add these figures in Supplementary Information (Supplementary Fig. 1) and discussion in the manuscript (para2, page 8).

Overall calculations domains in the simulations are quite small, with rather coarse discretisation (few grid points). Given the diffuse interface in the CH/PFM used, I wonder how much bulk is present in the phases, or whether the microstructure consist mainly of interface region.

We thank the Reviewer for pointing out this important aspect of CH/phase field model. In our model, the gradual change of the composition is possible, even if the interface is diffused. The interfacial region can be clearly identified from the spatial profile of the gradient of the free energy shown above (previous comment). The interfacial thickness is related to the (visible) non-zero gradient region. This region is very thin and hence we can expect bulk regions even in the smallest precipitates of the microstructures. We add this discussion in the manuscript (para2, page 8). Moreover, when we compared our microstructure in Ni 82 % (Fig. 5(r)), with the experimental one in ref. 18 in terms of the size, we approximated that our grid spacing, $\Delta x \approx 0.03 \mu\text{m}$ implying the total size of each plots to be $3.7 \times 3.7 \mu\text{m}^2$, which we believe is large enough to observe microstructures. We have performed simulation for one of the composition (Ni 60%) with a larger grid points (200x200) and shown the microstructure below:

Region VII in Figure 5 shows an inversion of the microstructure from Y' in Y. Find answers in product info, Q&As, reviews matrix to Y in Y' matrix. This is experimentally not commonly observed.

We totally agree with the Reviewer. Our results show that in Region VII, pure Ni and Ni₃Al are mixed. At relatively high Ni concentration, pure Ni precipitates in the Ni₃Al matrix, while at relatively low Ni concentration, Ni₃Al precipitates in

the pure Ni matrix. We expect this should be so because otherwise, the average concentration cannot be divided into the large precipitate region and small matrix region when the precipitation grows up. Therefore, we think our results are correct at least theoretically.

It is true that there is not yet any experimental demonstration of microstructures having Ni precipitates in Ni₃Al matrix, commonly termed as inverse superalloy. However, there is indication of possibility of such microstructure in some experimental papers as these microstructures are expected to have better hardness compared to the normal alloys (γ' precipitates in γ matrix). Vogel et. al. (Nat. Comm. 4:2955 (2013)) showed γ' -Ni particles in the γ' -Ni₃Al precipitates in hierarchical microstructures of Ni-Al alloy. A very recent paper (Vogel. F. et al., Acta Materialia, 157, 326 (2018)) established an analogy between the inverse alloy and the hierarchical Ni_{86.1}Al_{8.5}Ti_{5.4} alloy. Therefore, we believe that the prediction of such an inverse alloy will be seen in experiments too. However, this is a very important comment, so that we add a short explanation about the problem raised by the Reviewer in the revised manuscript (para 2, page 9).

Sensitivity of the model to changes in alloy composition does not seem very high, as phase fraction do not seem to differ very much between results. As the authors stress out repeatedly that the model aims at supporting alloy development from ab-initio, this does matter.

We agree with the Reviewer that in the narrow region of the phase diagram the sensitivity of the model is not very good. However, we want to bring it to the notice that in wide regions the microstructure is very sensitive to the composition. For example, in region VII, an 1% increase in Ni concentration lead to increase in precipitate size as well as increase in number of particle, and hence eventually change in phase fraction. Moreover we can see an inverse composition of precipitates and matrix after a slight increase in Ni % (between Ni 80% and Ni 82%). Similar changes are also seen in other regions too.

Finally, the author claim repeatedly that the model has no empirical parameters. The chemical potential contains however the gradient energy term, the coefficient of which determined interface thickness and - more importantly - interfacial energy. Equation 2 contains a mobility coefficient M_x , which can be linked to the diffusion coefficient (through the atomistic mobility). These parameters were NOT determined from the ab-initio calculations, but chosen at random. Characteristic length scale, growth kinetics and resulting microstructure critically depend on these parameters. I do therefore NOT agree that there are no empirical parameters in the model. The authors should at least be more specific in their claim and state that the thermodynamic part of the model does not contain empirical parameters.

We totally agree with this Reviewer's comment and we change the original words to "the thermodynamic part of the model does not contain empirical

parameters." We now removed the word "any" before "empirical parameter" from the title. We thank the Reviewer for correcting our original statement.

Finally, there have been a string of publication from T Mohri, dating as far back as 2004, combining phase-field with CVM calculations, producing microstructures from ab-initio. I would like the author to comment in their paper how their approach is different and novel.

We followed the suggestion by the Reviewer and now add proper reference of T Mohri's work on the phase-field method using first-principles CVM calculation, which we have shortly noted in the original manuscript (para 2, page 2). In their paper, T Mohri fixed the composition and treated the order-disorder phase transformation at different temperatures, which was of course novel at that time. However, one weak point is, as we have already written in our original manuscript, they cannot discuss the overall region of the phase diagram in their method. On the other hand, our method can be obviously applicable to any solid state region of the phase diagram. For example, if we change the temperature, we can obtain the microstructures at different temperatures as shown below for Ni 60% composition:

With increase in temperature the bright phase is decreasing slightly (Supplementary Fig. 2).

Generally, the paper is well written and readable and contains interesting thoughts.

We thank the Reviewer for his/her valuable comments and suggestion which helped us in improving the manuscript. We have now answered all the queries, comments and suggestions by the Reviewer and modified the manuscript accordingly. We hope the reviewer finds the current version acceptable for the publication.

Reviewers' comments:

Reviewer #1 (Remarks to the Author):

The authors have properly responded to the criticism drawn by the reviewers and the revised version of the manuscript is significantly improved. The problem with this manuscript is that the authors admitted (a few times) in their response to referees that their simulations do contain empirical input - but then the title of the manuscript is misleading. And some parts of the manuscript are misleading, too. Dropping the word "any" does not solve this problem. My suggestion is that the authors reformulate the current (misleading) title to something like "...predicting multi-composition phase separation with minimum empirical input" and re-formulate the rest of the manuscript in a similar manner. The manuscript altered in this way will be suitable for publication.

Reviewer #2 (Remarks to the Author):

Thank you for taking the reviewers comments on board in a constructive manner. I am fine with most modifications made, except a few minor issues.

Page 2: Thank you for acknowledging the previous works on CVM-Phase-field. Please refine the last sentence of the added text. What do you mean by 'region of the phase diagram.'? To me this refers to a single or two phase region, but I suspect you mean 'the entire phase diagram'.

Page 8/9: I am not entirely satisfied with your explanation why you see cuboid precipitates. A standard Cahn-Hilliard model for Ni-Al will give spherical particles, unless coherency stresses are included in the model, or there are numerical problems with the model. Your explanation uses interfacial stress, which is just standard interfacial free energy, as explanation. Why does that give cuboidal precipitates?

Some of the corrections appear to be made in a hurry, and would benefit from careful proof reading, to be brought in line with the quality of language of the rest of the paper.

Our Second Reply to Reviewer #1's Comments

Reviewer #1 (Remarks to the Author):

The authors have properly responded to the criticism drawn by the reviewers and the revised version of the manuscript is significantly improved. The problem with this manuscript is that the authors admitted (a few times) in their response to referees that their simulations do contain empirical input - but then the title of the manuscript is misleading. And some parts of the manuscript are misleading, too. Dropping the word "any" does not solve this problem. My suggestion is that the authors reformulate the current (misleading) title to something like "...predicting multi-composition phase separation with minimum empirical input" and reformulate the rest of the manuscript in a similar manner. The manuscript altered in this way will be suitable for publication.

Our Reply: We thank the reviewer for reviewing our manuscript again and approving most of the modifications, we made in our previous reply.

Concerning the empirical parameter: The reviewer is right that in our previous response to Reviewer 2, we wrote that our simulations do contain scaling parameters for the spatial length and the time step. However, we observed that, these parameters do not affect the final microstructures at all. What we assumed are not the empirical parameters but can be arbitrary numbers by which the length and time scales are determined. Since the time scale and the length scale are arbitrary, this has a particular importance. That is, the resulting pattern is universal in the scale. In the previous revision, we obeyed Reviewer 2's suggestion that we should at least be more specific in our claim and state that the thermodynamic part of the model does not contain empirical parameters. Reviewer 2 now agrees with our modification of the title and texts. We understand the concern of the reviewer regarding this comment, however, we want to preserve the words "without empirical parameter" in the title, because this is the key word of our manuscript. So, we want to propose to change the title from the previous "without ~~any~~ empirical parameter." to "without **thermodynamic** empirical parameter." Moreover, we will delete "any" from the related sentence in the Abstract "without ~~any~~ empirical parameter in the thermodynamic part." and completely delete the sentence "**Therefore, there is no empirical parameter in our model.**" in Page 5. We sincerely hope that this second revision would receive the approval of publication from you. Thank you very much once again for your very kind and constructive review.

We sincerely hope that this second revision would receive the approval of publication from the Reviewer. We thank the Reviewer very much once again for his/her very kind and constructive review.

We prepared a Comparable Manuscript showing our first revision in red with new modifications using red annotations for easy looking at which parts were modified from our original and previous (first revised) manuscripts.

Our Second Reply to Reviewer #2's Comments

Reviewer #2 (Remarks to the Author):

Thank you for taking the reviewers comments on board in a constructive manner. I am fine with most modifications made, except a few minor issues.

Our Reply: We thank the Reviewer for reviewing our manuscript again and approving most of the modifications we made in our previous reply.

Page 2: Thank you for acknowledging the previous works on CVM-Phase-field. Please refine the last sentence of the added text. What do you mean by 'region of the phase diagram.'? To me this refers to a single or two phase region, but I suspect you mean 'the entire phase diagram'.

Our Reply: According to the Reviewer's suggestion, we will change “the overall region of the phase diagram.” to “the entire phase diagram.”

Page 8/9: I am not entirely satisfied with your explanation why you see cuboidal precipitates. A standard Cahn-Hilliard model for Ni-Al will give spherical particles, unless coherency stresses are included in the model, or there are numerical problems with the model. Your explanation uses interfacial stress, which is just standard interfacial free energy, as explanation. Why does that give cuboidal precipitates?

Our Reply: In the case of Ni 25.5% alloy as shown in Figure 5(d), for example, we obtained round shape precipitates. Therefore, we do not always have cuboidal precipitates in our PFM. We have observed that, the shape of the precipitates depends on the sharpness of the interface boundary. There is a tendency to form cuboidal precipitates when the interfaces are sharp, while round precipitates appear for diffuse interfaces. In atomic scale, the lattice mismatch is the basic origin of the coherency strain. The difference in the lattice parameter can be included in our PFM as vacancies and interstitials as well as the mixture of the two phases. Tensile and compressive strain corresponds to high and low number density, which can be viewed as the clusters with interstitial and the vacancy clusters. Thus, the effect of lattice mismatch and hence the coherency strain can be included in our PFM in which the free energy is a discrete variable for integer compositions. This is not possible in all previous PFMs in which the free energy is a continuous variable with respect to compositions. This is a distinct advantage of the present PFM. The most important point is that, in our PFM, the density is automatically fine tuned by the clusters with vacancies and interstitials as well as the mixture of the phases. Therefore, the detailed lattice mismatch can be handled without introducing anisotropic elastic energies. If the interfacial thickness is as small as the size of the fine mesh of the simulation cell, the effect of lattice mismatch becomes important, and cuboidal precipitates appear. On the other hand, if the interface is broad and diffuse, the effect of lattice mismatch becomes less important, and round precipitates appear. Thus, we can conclude that the main origin of the cuboidal precipitates is that our free energy is a discrete variable. Indeed, if we change the definition of the free energy to be a continuous variable with respect to compositions, we obtain round precipitates. (See the figures below). Therefore, this is not at all a numerical problem of our model. We will add this explanation (the sentences written above in green color) in the text.

Microstructures obtained for Ni 65% alloy for free energy being a continuous function for the compositions $n+m \geq 4$ and discrete for the clusters with vacancy

Some of the corrections appear to be made in a hurry, and would benefit from careful proof reading, to be brought in line with the quality of language of the rest of the paper.

Our Reply: We will change the followings:

“A CVM based phase field method has been developed,”
→ “A CVM based phase field method has been **also** developed,”
and “In this method the alloy composition was fixed”
→ “In this method the alloy composition was **kept** fixed” in Page 2

“while for **the** non-conserved order parameter,”
→ “while for non-conserved order parameters,” in Page 5

“The plot of the total stress of the system **with** simulation step”
→ “The plot of the total stress of the system **versus** the simulation step” in the bottom line of Page 8

“After this the stress decreases”
→ “After this **increase**, the stress **slightly** decreases”
and “This initial increase in stress” → “This initial increase in **the** stress” in Page 9

Also, we will put comma (,) in between “After the system forms a stable microstructure” and “the stress also becomes constant.” in the next line of Page 9.

We sincerely hope that this second revision would receive the approval of publication from the Reviewer. We thank the Reviewer very much once again for his/her very kind and constructive review.

We prepared a Comparable Manuscript showing our first revision in red with new modifications using red annotations for easy looking at which parts were modified from our original and previous (first revised) manuscripts.

REVIEWERS' COMMENTS:

Reviewer#1

I have checked the latest version of the manuscript as well as authors' response to Reviewer #2 and Reviewer's critical comments. Well, I would accept the manuscript. The crucial point (as identified by Reviewer #2 but it is my opinion, too) is the following statement:

=====
=====

Thus, we can conclude that the main origin of the cuboidal precipitates is that our free energy is a discrete variable.

=====
=====

This conclusion goes against "traditional explanations" but the authors could formulate it only because their computational method is unique. So, I think that this particular conclusion deserves a very broad discussion and your journal is just the best platform to start it.

I am afraid that only time will show who is right here (if classics from 1960's or the authors of this manuscript) but this paper will certainly be cited and has a significant potential for the whole field. So, I suggest to accept it.

Our Third Reply to Reviewer #1's Comments

REVIEWERS' COMMENTS:

Reviewer#1

I have checked the latest version of the manuscript as well as authors' response to Reviewer #2 and Reviewer's critical comments. Well, I would accept the manuscript. The crucial point (as identified by Reviewer #2 but it is my opinion, too) is the following statement:

=====
=====

Thus, we can conclude that the main origin of the cuboidal precipitates is that our free energy is a discrete variable.

=====
=====

This conclusion goes against traditional explanations but the authors could formulate it only because their computational method is unique. So, I think that this particular conclusion deserves a very broad discussion and your journal is just the best platform to start it.

I am afraid that only time will show who is right here (if classics from 1960's or the authors of this manuscript) but this paper will certainly be cited and has a significant potential for the whole field. So, I suggest to accept it.

Our Reply: We are grateful to the Reviewer for critically reviewing our modified manuscript and the authors' response to the Reviewers. The constructive comments of the Reviewer have improved the manuscript significantly. We are very glad to know that the Reviewer have accepted the last modified manuscript for publication. We thank the Reviewer for finding our computation method unique and giving it a chance for an exposure to the research community through Nature Communication for a broader discussion in the future.

Since there is no modification suggestions from the Reviewer, we have modified the manuscript as per the editor's requests. We prepared a Comparable Manuscript showing our latest revision in red for easy looking at which parts were modified from our previous version of the manuscript.